The unnatural history of Kāne‘ohe Bay: coral reef resilience in the face of centuries of anthropogenic impacts

Bahr Keisha D. kbahr@hawaii.edu
Jokiel Paul L.
Toonen Robert J.
University of Hawaiʻi, Hawaiʻi Institute of Marine Biology , Kāneʻohe, HI , USA
Bruno John
Electronic publication date: 2015 May 12
Publication date: 2015
Volume: 3
Electronic Location ID: e950
Received 2015 Jan 4; Accepted 2015 Apr 20
Copyright: © 2015 Bahr et al.
Copyright year: 2015
Copyright holder: Bahr et al.
License: This is an open access article distributed under the terms of the Creative Commons Attribution License, which permits unrestricted use, distribution, reproduction and adaptation in any medium and for any purpose provided that it is properly attributed. For attribution, the original author(s), title, publication source (PeerJ) and either DOI or URL of the article must be cited.
License URL: https://creativecommons.org/licenses/by/4.0/

Keywords: Reef resilience, Hawaii, Climate change, Coral reefs, Kāneʻohe Bay, Corals, Natural history, Eutrophication

Funding: The United States Geological Survey Pacific Coastal and Marine Science Center G13AC00130 The Hawai’i Institute of Marine Biology (HIMB) contribution 1622 School of Ocean and Eart Science and Technology (SOEST) contribution 9325 Funding for this work was provided by the Colonel Willys E. Lord & Sandina L. Lord Endowed Scholarship and the Charles H. and Margaret B. Edmondson Research Fund. This work is also partially supported by the United States Geological Survey Pacific Coastal and Marine Science Center Cooperative Agreement G13AC00130. This is the Hawai’i Institute of Marine Biology (HIMB) contribution #1622 and the School of Ocean and Eart Science and Technology (SOEST) contribution #9325. The funders had no role in study design, data collection and analysis, decision to publish, or preparation of the manuscript.

==============================
Kāneʻohe Bay, which is located on the on the NE coast of Oʻahu, Hawaiʻi, represents one of the most intensively studied estuarine coral reef ecosystems in the world. Despite a long history of anthropogenic disturbance, from early settlement to post European contact, the coral reef ecosystem of Kāneʻohe Bay appears to be in better condition in comparison to other reefs around the world. The island of Moku o Loʻe (Coconut Island) in the southern region of the bay became home to the Hawaiʻi Institute of Marine Biology in 1947, where researchers have since documented the various aspects of the unique physical, chemical, and biological features of this coral reef ecosystem. The first human contact by voyaging Polynesians occurred at least 700 years ago. By A.D. 1250 Polynesians voyagers had settled inhabitable islands in the region which led to development of an intensive agricultural, fish pond and ocean resource system that supported a large human population. Anthropogenic disturbance initially involved clearing of land for agriculture, intentional or accidental introduction of alien species, modification of streams to supply water for taro culture, and construction of massive shoreline fish pond enclosures and extensive terraces in the valleys that were used for taro culture. The arrival by the first Europeans in 1778 led to further introductions of plants and animals that radically changed the landscape. Subsequent development of a plantation agricultural system led to increased human immigration, population growth and an end to traditional land and water management practices. The reefs were devastated by extensive dredge and fill operations as well as rapid growth of human population, which led to extensive urbanization of the watershed. By the 1960’s the bay was severely impacted by increased sewage discharge along with increased sedimentation due to improper grading practices and stream channelization, resulting in extensive loss of coral cover. The reefs of Kāneʻohe Bay developed under estuarine conditions and thus have been subjected to multiple natural stresses. These include storm floods, a more extreme temperature range than more oceanic reefs, high rates of sedimentation, and exposure at extreme low tides. Deposition and degradation of organic materials carried into the bay from the watershed results in low pH conditions such that according to some ocean acidification projections the rich coral reefs in the bay should not exist. Increased global temperature due to anthropogenic fossil fuel emmisions is now impacting these reefs with the first “bleaching event” in 1996 and a second more severe event in 2014. The reefs of Kāneʻohe Bay have developed and persist under rather severe natural and anthropogenic perturbations. To date, these reefs have proved to be very resilient once the stressor has been removed. A major question remains to be answered concerning the limits of Kāneʻohe Bay reef resilience in the face of global climate change.

Introduction

The Kāne‘ohe Bay ecosystem is located on the northeast coast of Oʻahu, Hawaiʻi (21°, 28′N; 157° 48′W) (Fig. 1), and consists of the watershed, the semi-enclosed embayment, and the near shore oceanic environment. Kāneʻohe Bay is the largest sheltered body of water in the main eight Hawaiian Islands with total surface area of 41.4 km2 at mean surface levels (Jokiel, 1991). Along the southwest to southeast axis, the bay is approximately 4.3 km wide with a length of 12.8 km and an average depth of 10 m (Smith, Chave & Kam, 1973; Jokiel, 1991). The bay is bounded by a barrier reef on the seaward side that is cut by two major channels. The lagoon formed by these features is continually flushed by oceanic waves driving over the barrier reef and by tidal change through the channels. The bathymetry of the bay is divided into the inshore and offshore portions (Bathen, 1968). The offshore portion (34%) consists almost entirely of extensive shallow coral and sand reef 0.3 m–1.2 m in depth. The inshore portion comprises 66% of the total area, and is characterized by the estuarine lagoon, which holds numerous patch reefs that occur at depths of less than 1 m from the surface, and are partially exposed during extreme spring tides (Jokiel, 1991). The lagoon is generally divided into three sectors (southeast, central and northwest) based on the circulation and relative degree of oceanic influence (Bathen, 1968; Smith et al., 1981). The entire shoreline, except parts of Mokapu Peninsula, is ringed by shallow fringing reef. The deepest portion of the bay is 19 m, and the substratum is predominately coral rubble, gray coral mud, and fine coral sands throughout (detailed in Supplemental Information; Jokiel, 1991).

Figure 1 Dredging and filling areas in Kāneʻohe Bay, Oʻahu Hawaiʻi.

Dredged areas (red) and filled areas (black) in Kāneʻohe Bay on the island of Oʻahu. Modified after Maragos 1972. Photo Credit: Quickbird Digital Globe.

The Kāneʻohe Bay ecosystem has an intriguing history of anthropogenic influence beginning with colonization by Polynesians circa 1250 A.D. (Kittinger et al., 2011). Western contact during the eighteenth century was followed by extensive plantation agriculture, land modification, introduction of numerous species, and eventually urbanization (Devaney et al., 1982). Further, the bay is currently showing initial indicators of climate change with major bleaching events in 1996 (Jokiel & Brown, 2004) and again in 2014 (Neilson, 2014).

Due to restricted circulation, the temperatures in this bay are historically 1–2 °C higher than the open ocean in the summer months. Consequently, the corals in the bay are already living at temperatures that offshore ocean reefs will not experience for many years under various scenarios of global warming. Likewise, the corals are already living at elevated pCO2. Fagan & Mackenzie (2007) found that pCO2 was approximately 500 µatm on average in the northern bay while central and southern bay waters had an average pCO2 of 460 µatm with the entire bay and nearshore reef experiencing levels well above atmospheric pCO2 (Shamberger et al., 2011). Such levels of pCO2 are believed to be highly deleterious to coral growth (summarized by Hoegh-Guldberg et al., 2007). One estimate is that when atmospheric partial pressure of CO2 reaches 560 µatm all coral reefs will cease to grow and start to dissolve (Silverman et al., 2009). Nevertheless, rich coral reefs exist in Kāneʻohe Bay at levels of temperature and pCO2 that will not be experienced in oceanic waters until later in the century.

Reefs throughout the world have undergone and are undergoing change. Kāneʻohe Bay is unique because the changes have been documented in detail by scientists at the Hawaiʻi Institute of Marine Biology and others with more than 1600 peer-reviewed publications over a half-century of continuous research (Kāneʻohe Bay Information System, 2014). As such, these reefs provide a fascinating model system to study the effects of global climate change on coral reefs at greatest risk around the planet and those in close proximity to urbanized regions around the globe. In this review, we highlight the value of this exemplary ecosystem through a description of reef resilience demonstrated by recovery from repeated anthropogenic and natural disturbances. The limits of Kāneʻohe Bay reef resilience in the face of global climate change remains a major question and will be a focus for future research.

Historical Impact of Human Population Growth in the “Polynesia Era” and the “Western Era”

Polynesian era

Initial impact of the first Polynesian settlers included the introduction of “canoe plants” (e.g., the coconut, yam, breadfruit, taro, etc.) as well as domestic animals (e.g., pigs, chickens, dogs, etc.) along with accidental introductions. As human population increased, changes occurred in the watershed and adjacent bay environment. Depletion of nearshore fish stocks (Kittinger et al., 2011) led to increased development of agricultural resources and construction of fishponds. A total of thirty fishponds covering up to 30% of the shoreline were estimated to exist in Kāneʻohe Bay during the 19th-century (Jackson, 1882; Devaney et al., 1982). The construction of terraced (loʻi kalo) systems collected stream runoff, which was used as irrigation water for taro patches and cultivation of other plants. This extensive system of combined agriculture and aquaculture coexisted without extreme degredation of the natural environment for over 750 years (Kittinger et al., 2011) and is believed to have supported a population in Kāneʻohe larger than that of the present day which is approximately 35,000 according to United States Census Bureau (http://www.census.gov).

Western era

Arrival of the first Europeans led to introduction of human diseases not known previously in the islands leading to population decline throughout the Hawaiian kingdom (Stannard, 1989). The magnitude and timing of the decline is a matter of debate, but Bushnell (1993) concludes: “From a historian’s perspective this demographic collapse, continuing as it did throughout the nineteenth century, is the most important “fact” in Hawaiian history. As disease destroyed their numbers, it destroyed the people’s confidence and their culture; finally, it was the most important factor in their dispossession: the loss of their land and ultimately of their independence. Consider how different the fate of Hawaiʻi would have been if the numbers of Hawaiians had remained undiminished from what they had been in 1778, whether those numbers were 300,000 or 400,000 or more—instead of the fewer than 40,000 who remained alive in 1893.” Private ownership of land was instituted during the Great Mahele of 1848 which led to a plantation agricultural system with many acres of land devoted to cultivation of a single crop (sugar or pineapple), which was harvested and exported for profit (Bates, 1854; Devaney et al., 1982). The area of grasslands for livestock in the area increased from 700 acres in the 1880s to 3,000 acres in 1969 (Maragos & Chave, 1973; Devaney et al., 1982). Livestock grazing is one of several causes of deforestation leading to erosion and increased sediment loading in the streams, with consequent negative influences on the natural terrestrial, aquatic and marine biotic environment (Devaney et al., 1982).

Major Impacts During the Western Era

Prior to 1930, the coral reefs of Kāneʻohe Bay were still in excellent condition. For example, the area of the south basin subsequently impacted by dredging, sedimentation and sewage discharge was described as the ‘coral garden’ (MacKay, 1915; Edmondson, 1929; Devaney et al., 1976; Hunter & Evans, 1995). The commercial enterprise Coral Gardens began glass-bottom boat tours prior to 1915 to show the beautiful underwater reefs in the south bay. Coral Gardens remained a well-known tourist attraction but ceased operations just prior to World War II when the area was dredged for seaplane runways (Devaney et al., 1976), and has never recovered its former condition.

Dredging and filling

Prior to 1939, dredging was limited to small areas around boat landings and piers. During the construction of what was then known as the Kāneʻohe Bay Naval Air Station on Mokapu Peninsula (1939–1945) extensive dredging occurred throughout the bay (Fig. 1). Physical changes in the bay included alterations to the shoreline ranging from 5% of the northern bay, 68% of the central bay, to 88% of the southern bay (Hunter, 1993) (Fig. 1). A deeper ship channel was dredged 1.5 km west of Mokapu to extend the entire length of the bay. This ship channel intersects the Sampan Channel (Kāneʻohe Passage), which has a natural depth of 2.4 m, to allow access at the southeast portion of the bay for smaller ships. At the end of the ship channel is the Mokoliʻi passage, which was dredged to 7.6 m to allow access for larger ships between the bay and the open ocean (Jokiel, 1991). Also, 25 of the 79 patch reefs (5% of total patch reef area) were partially or entirely dredged during this period (Devaney et al., 1976; Hunter & Evans, 1995). Based on available records, the total estimated dredged material was 11,616, 300 m3 (Jokiel, 1991). Lack of detailed records precludes a complete understanding of the amount of material removed and dumped into other areas of the bay, but the volume was clearly extensive and is believed to be responsible for shoaling (Roy, 1970; Devaney et al., 1982). Forty-five years after the first major hydrographic survey (Jackson, 1882), the Coast and Geodetic Survey revealed no significant changes in the depth or bathymetry of the lagoon between 1882 and 1927 (Roy, 1970). By comparison, a fathometer survey showed an average decrease of 1.7 m in depth of the lagoon over the 42 years following the 1927 survey (Roy, 1970). Recovery of corals has not occurred in deeper dredged areas; however, some recovery occurred on the reef slopes of the dredging areas in shallow waters (Maragos, 1972). Corals in the south bay have not recovered, particularly in areas of soft substratum or in areas of high sedimentation that impedes recruitment and essential irradiance for coral growth (Bosch, 1967; Maragos, 1972; Jokiel, 1991). Clearly, the effects of dredging on these coral reef communities are long term. Additionally, nine fishponds (total area of 0.24 km2) were filled for land development between 1946 and 1948, three additional ponds (total area of 0.03 km2) were filled since the 1950s, and others have been partly destroyed or altered. As a result, only 12 of the 30 walled fishponds that once existed in Kāneʻohe Bay remain. Only three of these remain intact, and most of the others have fallen into such disrepair that they are unknown to local residents and not immediately recognizable as fishponds (Devaney et al., 1982; Farber, 1997).

Land management and sedimentation

Land runoff and sedimentation due to rapid urbanization have increased dramatically in Kāneʻohe Bay since 1940 with adverse effects on the bay environment. Human population increased by 130% between 1940 and 1950 and 190% between 1950 and 1960 (Devaney et al., 1982; Smith, Chave & Kam, 1973; Jokiel, 1991). During this period of rapid urbanization, the grading of land resulted in increased sedimentation and increased sewage discharge (Banner, 1974). Roughly 70% of the sediment in the bay is internally derived from the dredging and breakdown of calcium carbonate materials, with the remainder of sediment coming from terrestrial run-off (Roy, 1970). The organic material from terrigenous input in the sediments negatively influences coral settlement, species richness and diversity (Friedlander et al., 2008).

Sewage

Before 1963, Kāneʻohe was served by private septic tanks and cesspools that discharged into the ground water and eventually into Kāneʻohe Stream, entering the southern corner of Kāneʻohe Bay. In 1963, a secondary treatment plant was built with an outfall at 8 m water depth in the southeast corner of the south basin. Another outfall from the Kāneʻohe Marine Corps Air Station discharged into the northeast corner of the south basin. Later in 1970, a small secondary sewage treatment plant was built and discharged into the northwestern portion of the bay (Devaney et al., 1982; Jokiel, 1991). During this period, oxygen levels in the south bay were low due to the oxidation of organic material, which created respiratory stress and promoted anoxic conditions. Moreover, reduction in light penetration from increased productivity in the water column negatively affected the survival and growth of corals. The sewage effluent discharge decreased species diversity, increased eutrophication, and substantially altered ecosystem structure away from a coral dominated ecosystem (Pastorok & Bilyard, 1985; Jokiel, 1991; Stimson, Larned & Conklin, 2001).

In 1973, Kāneʻohe Bay was described as “a reef ecosystem under stress” (Smith, Chave & Kam, 1973) due to rapid and large population increase within the surrounding areas as well as severe impacts from extensive dredging, increased sedimentation, stream channelization, and municipal sewage discharge within the previous 30-year period (Hunter & Evans, 1995). The high coral cover once recorded in the southern bay declined quickly with these heavy alterations and major sewage spills in December 1977 and May 1978. The sewage outfall was moved from Kāneʻohe Bay and diverted to deeper waters off Mokapu Peninsula in 1979 after nearly 20 years of continuous discharge into the bay (Smith et al., 1981; Jokiel, 1991).

The sewage diversion led to a dramatic decrease in nutrient levels, turbidity and phytoplankton abundance (Smith et al., 1981). These conditions led to rapid recovery in coral reef populations in the south bay over a relatively short time period (Hunter & Evans, 1995). Before and after sewage diversion, studies indicate that the primary benthic response to nutrient loading was a large buildup of plankton biomass, which supported a benthic community dominated by filter and deposit feeders. The cycling among the heterotrophs, autotrophs, detritus, and inorganic nutrients drove the biological communities (Hunter & Evans, 1995). The high nutrient levels supported rapid growth and abundance of the native alga, Dictyosphaeria cavernosa, in the bay. Early surveys (Banner, 1974) revealed that D. cavernosa mats overgrew and eliminated corals (Fig. 2). The “phase shift” from corals to algae has been attributed to nutrient enrichment resulting from sewage discharge (Pastorok & Bilyard, 1985; Stimson, Larned & Conklin, 2001). After sewage diversion in 1979, the plankton and the benthic biomass decreased rapidly and the corals gradually began recovery to pre-sewage condition (Smith et al., 1981). Long-term trends in recovery of corals and algal abundance in Kāneʻohe Bay after the 1979 sewage diversion were assessed at fifteen sites originally established in 1970–71 (Maragos, 1972) and resurveyed in 1983 (Maragos, Evans & Holthus, 1985; Evans, Maragos & Holthus, 1986) and again in 1990 (Hunter & Evans, 1995). Surveys from 1971 to 1983 showed that coral cover more than doubled from 12% in 1971 to 26% in 1983 accompanied by a decline in the abundance of the coverage of D. cavernosa (Hunter & Evans, 1995). The 1990 surveys revealed that the rate of coral recovery had slowed and even reversed at some sites (Hunter & Evans, 1995).

Figure 2 Algal dominance.

Photographs of Dictyospheria cavernosa over growth of Porites compressa colony at a long term monitoring site in Kāneʻohe Bay in 1999 and 2000. Photographs by PL Jokiel.

By 1984 the coverage of D. cavernosa had decreased in the south sector of Kāneʻohe Bay, but remained relatively high in the central bay through the 1990s (Stimson, Larned & Conklin, 2001). The persistence of D. cavernosa in this region was attributed to overfishing that reduced grazing pressure on D. cavernosa by herbivorous fish (Stimson, Larned & Conklin, 2001). Dictyosphaeria cavernosa declined dramatically throughout the bay between February and June 2006 (Stimson & Conklin, 2008) as the result of very low irradiance levels during an unusual 42-day period of rain and heavy overcast in March 2006. There has been no resurgence of the alga since 2008 decline. This is a rare example of a reverse phase shift from algal domination to a coral dominated reef community (Stimson & Conklin, 2008). Today Kāneʻohe Bay stands out as among the better reef sites across the Main Hawaiian Islands (Rodgers et al., 2010).

Fishing pressure

Overfishing has been a concern in Kāneʻohe Bay since ancient times, and the construction of 30 fishponds in the bay was one method that the Polynesian inhabitants used to increased fish production. Over a period of many centuries, the early inhabitants learned to offset fishing pressure through development of a carefully regulated and sustainable “ahupua‘a” management system that integrated watershed, freshwater and nearshore marine resources based on the fundamental linkages between all ecosystems from the mountain tops to the sea (Lowe, 2004; Jokiel et al., 2011). This traditional scheme employed adaptive management practices keyed to detection of subtle changes in natural resources. Sophisticated social controls on resource utilization were an important component of the system. Over the past two centuries, a Western management system has gradually replaced much of the traditional Hawaiian system. There are major differences between the two systems in the areas of management practices, management focus, knowledge base, dissemination of information, resource monitoring, legal authority, access rights, stewardship and enforcement (Jokiel et al., 2011). Even though a much smaller proportion of the human population presently fishes or consumes local fish products relative to ancient times, marine resources have steadily declined over time coincident with the shift away from the traditional Hawaiian management system (Lowe, 2004; Jokiel et al., 2011; Kittinger et al., 2011). However, there has been a recent shift toward incorporating elements of the traditional scheme using methods and terminology acceptable and appropriate to present day realities.

Trends in reported landings, trips, and catch per unit effort for Hawaiʻi fisheries have been reported by Smith (1993). In heavily populated areas such as Kāneʻohe Bay, fishing pressure appears to exceed the capacity of inshore resources to renew themselves. The current fisheries of Kāneʻohe Bay have been characterized as multispecies, multigear with relatively low yields that suggest overfishing (Everson & Friedlander, 2004). Hook and line fishing was the dominant method accounting for 55% of the active fishing effort in Kāneʻohe Bay during a 1991–1992 survey, but today gill and surround nets account for the majority of the fish catch (Everson & Friedlander, 2004). Yield per area figures for Kāneʻohe (0.92–1.4 t km−2 yr−1) for the entire catch and 0.8 t km−2 yr−1 (excluding small coastal pelagic species) are similar but lower compared with estimates from other coral reef habitats throughout Hawaiʻi and the Indo-Pacific. The high effort and low catch per unit effort (CPUE) for pole-and-line fishers has been cited as evidence that species targeted by this method are being overfished. According to Everson & Friedlander (2004), these fishers were the most vocal in proclaiming that the fishery resources of the bay were in serious decline. Many of the species targeted by pole-and-line fishers are also caught by gill netters, who are responsible for the majority of the catch.

Brock, Lewis & Wass (1979) documented the resilience of Kāneʻohe Bay coral reef fish populations to extreme perturbation. All fishes residing on a small isolated coral patch reef with an area ∼1,500 m2 were collected in 1966 followed by another collection on the same reef in 1977. The patch reef was surrounded by nets and all fishes were killed and collected using rotenone. The assemblages of fishes from the two time periods were similar in trophic structure and standing crop. Planktivorous fishes were the most important trophic group as the result of the abundant zooplankton food resources in the lagoon. After the second collection in 1977, recolonization by fishes was monitored for 1 year. Recolonization proceeded rapidly, primarily by juvenile fishes that were well beyond the larval metamorphosis stage. Within 6 months of the 1977 collection, the trophic structure had again been re-established. The MacArthur-Wilson model (MacArthur & Wilson, 1967) of insular colonization adequately described the recolonization process with an equilibrium situation being reached in less than 2 years. A relatively deterministic pattern of recruitment emerged. The fish populations proved to be a persistent and predictable entity. These data indicate that fish communities in Kāneʻohe Bay are extremely resilient to extreme perturbations, and predict that fish populations will rebound rapidly whenever and wherever fishing pressure is reduced.

Introduced and invasive species

Invasive species pose a significant threat to the species diversity of the isolated, highly endemic coral reef communities in Hawaiʻi. Of the total biota in Kāneʻohe Bay, 14.5% are confirmed or assumed to be nonindigenous species (Coles, DeFelice & Eldredge, 2002; Friedlander et al., 2008). These nonindigenous species have a range of impacts where a majority are innocuous and find an unique niche in the coral reef community, while a few have a direct impact on the coral reef ecosystem.

A few nonindigenous invertebrates have displaced natives in the bay. For example, the Philippines stomatopod, Gonodactylaceus falcatus, has supplanted the native Pseudosquilla ciliate in the coral rubble habitats (Kinzie, 1968; Coles, DeFelice & Eldredge, 2002; Friedlander et al., 2008). The orange keyhole sponge (Mycale armata) is believed to be an unintentional introduction that was first described in Kāneʻohe Bay in 1996 (Coles et al., 2004) and is currently overgrowing corals and native sponge species (Friedlander et al., 2008).

Among the established introduced marine fishes (at least 13 species in Hawaiʻi), the introduced mullet (Valamugil engeli) and goatfish (Upeneus vittatus) are commonly found in the Kāneʻohe Bay (Randall, 1987; Friedlander et al., 2008). The introduced V. engeli, along with several species of introduced tilapia, are thought to affect the abundance of the native mullet (Mugil cephalus, ʻamaʻama) through competition for food and other resources (Randall, 1987; Eldredge, 1994). Also, the blacktail snapper (Lutjanus fulvus, toʻau) bluestripe snapper (Lutjanus kasmira, taʻape) and peacock grouper (Cephalopholis argus, roi) were intentionally introduced with the intent of supplementing stocks of edible reef fish (Gaither, Toonen & Bowen, 2012). Lutjanus kasmira (Blueline snapper) is regarded as an unfortunate introduction (Randall, 1987) due to its proliferation and lack of acceptance as a food fish by the public (Oda & Parrish, 1982). The species is very unpopular with fishermen who are convinced that its increase has been at the expense of more valuable species (Tabata, 1981). Cephalopholis argus is considered a serious predatory threat to native reef fishes (e.g., Zebrasoma flavescens) and is credited for the decline in aquarium fishes (Friedlander et al., 2008).

Since the 1950s, 19 species of seaweeds were intentionally or accidentally introduced into Hawaiʻi (Glenn & Doty, 1990). Among these, a few were successful in expanding their abundance and distribution in Kāneʻohe Bay. Avrainvillea amadelpha, a cryptogenic species, competes directly with other native species in particular the native seagreass (Halophila hawaiiana) and is hypothesized to have arrived after 1981 (Godwin, Rodgers & Jokiel, 2006). Acanthophora spicifera is believed to have been introduced accidentally from Guam and has spread throughout the state since the 1950s. This species can be found throughout Kāne’ohe Bay’s reefs specifically in shallow intertidal zones. The brittle branches often break off and grow rapidly, therefore proliferating and out-competing several species of native seaweeds (Russell & Balaz, 1994; Smith, Hunter & Smith, 2002). Gracilaria salicornia was introduced into the bay in 1970 evaluate its practicality as an aquaculture food product, and has since spread extensively, inhabiting the intertidal and subtidal up to about 4 m in depth (Smith et al., 2004). This seaweed is most prevalent on the fringing reefs, but can also be found on the barrier and patch reefs throughout the bay, growing in large, low, entangled clumps that over grow and kill reef-building corals (Smith et al., 2004; Godwin, Rodgers & Jokiel, 2006). Several Kappaphycus and Eucheuma species were also introduced into Kāneʻohe Bay in the 1970s for experimental aquaculture studies for the carrageenan industry (Doty, 1977; Pickering, Skelton & Sulu, 2007). Kappaphycus species have high morphological plasticity, which makes it hard to distinguish from Eucheuma species. Kappaphycus alvarezii was transported to Moku o Loʻe in 1974 and has successfully dispersed throughout Kāneʻohe Bay (Russell, 1983; Smith, Hunter & Smith, 2002). This species grows in strands that may extend over 1.8 m in length with diameteres up to 2.5 cm and can be found in abundance on the fringing reef and patch reefs in the central and southern sectors of the bay (Godwin, Rodgers & Jokiel, 2006). Since its introduction in 1974, surveys have revealed that Kappaphycus spp. has spread 9 km in 25 years, increased its distribution and have invaded new habitats (Rodgers & Cox, 1999; Conklin & Smith, 2005). These invasive species pose significant negative effects on the entire reef community by overgrowing, smothering and killing reef building corals and consequently stripping the reef of its complexity (Smith, 2003; Friedlander et al., 2008). Currently research, management and community eradication efforts are underway to control the distribution and limit abundance of these species (Godwin, Rodgers & Jokiel, 2006; Stimson, Cunha & Philippoff, 2007).

Historical Perturbations and Reef Resilience

Freshwater kills

Coral reefs in Kāneʻohe Bay periodically experience natural environmental disturbances from extreme low tides, freshwater input, and increased sea surface temperatures. Extreme low tides have exposed corals to desiccation, temperature changes, and fresh water from rainfall. Freshwater ‘kills’ are rare events that are caused by lowered salinity during severe storm flooding and runoff events (Banner, 1968; Coles & Jokiel, 1992; Jokiel et al., 1993) that modify the structure of reef communities. These events have been documented in Kāneʻohe Bay during May 1965 (Banner, 1968), again during January 1988 (Jokiel et al., 1993) and recently a less severe event occurred during flash floods in July 2014 (Bahr, Rodgers & Jokiel, 2015) for a frequency of re-occurrence of approximately 25 years. During the 1965 flood, the freshwater discharged within a 24 h period was calculated to be equivalent to a surface layer of 27 cm over the entire bay (Banner, 1968). Along with reduction in salinity, freshwater caused temperatures on the adjacent reef flat to decrease by 1–3 °C and average irradiance levels to decrease by 10–20% in the 1988 flood and by 55% in the more recent 2014 flood (Jokiel et al., 1993; Bahr, Rodgers & Jokiel, 2015). The reduction in salinity to 15‰–20‰ for a 24 h period or longer results in massive mortality of coral reef organisms in these shallow waters (Coles & Jokiel, 1992; Jokiel et al., 1993). Also, mass fish mortalities were documented in the 1965 flood, while later freshwater kill events did not produce massive mortalities of the fishes in the bay (Jokiel et al., 1993). The fish mortalities in the 1965 event probably resulted from anoxia caused by sewage discharge and low light penetration (Banner, 1974; Jokiel et al., 1993). Corals exposed to the reduced salinity events from the 1988 flooding event were observed to recover within five to ten years (Jokiel, 2008). Recovery from the 1965 flooding event was non-existent likely due to the persistent eutrophic conditions caused by municipal sewage discharge during that period.

Bleaching events

In the summer of 1996, offshore sea surface temperature was extremely high, and by early September, maximum mid-day temperatures on the reefs of Kāneʻohe Bay exceeded 30 °C. These conditions led to the first documented large scale bleaching event in the bay (Jokiel & Brown, 2004). Lack of wind-induced water motion decreased sediment resuspension and led to higher water transparency. Along with increased water clarity, prolonged lack of cloud cover increased solar input and resulted in further heating of the shallow inshore waters. Maximum bleaching in corals was observed in early September, 2–4 weeks after the peak of the highest temperatures (maximum 30.7 °C), in the inner portions of the bay. Coral species in the bay displayed variation in bleaching susceptibility. Porites evermanni, Cyphastrea ocellina, Fungia scutaria, and Porites brighami showed highest resistance to bleaching. Porites compressa, Porites lobata, Montipora capitata, and Montipora patula showed moderate resistance to bleaching. In contrast, Montipora flabellata, Pocillopora meandrina, Pocillopora damicornis, and Montipora dilatata showed high levels of susceptibility to bleaching (Jokiel & Brown, 2004). The corals recovered in the winter months with overall coral mortality of less than 2% (Jokiel & Brown, 2004).

In the summer of 2014, temperatures began to peak above 29 °C in mid-August, persisted for 6 weeks, and maintained mid-day maximum temperatures above 30 °C for a week in mid-September. These temperatures resulted in the second mass coral bleaching event for Kāneʻohe Bay (Neilson, 2014; K Bahr, 2014, unpublished data). This 2014 event was the most severe and extensive reported in the Hawaiian Archipelago to date, with evidence of coral bleaching reported from the Big Island of Hawaiʻi throughout the Archipelago extending to Midway Atoll (C Couch, pers. comm., 2014).

This recent bleaching event has been more severe (in terms of the proportion of colonies impacted) and has affected a much larger area in comparison to the 1996 bleaching event. Neilson (2014) estimates that 83% of the dominant corals species (e.g., Montipora capitata, Porites compressa, etc.) in the bay showed signs of thermal stress with partial or full loss of symbiont pigment. Bleaching intensity was variable throughout the bay; however, high levels of bleaching occurred on patch reefs in the north bay (Neilson, 2014). Also, bleaching intensity decreased with water depth and corals in areas of high turbidity were observed to suffer little to no bleaching at the same temperatures of corals in clearer waters. These observations reinforce the importance of irradiance in accelerating bleaching in corals (Jokiel & Brown, 2004). Recovery from the 2014 bleaching event was observed throughout the bay (Fig. 3); however, reefs that were influenced by the freshwater kill had lower levels of recovery and higher levels of mortality. Corals in the bay were slower to recover during the 2014 event (K Bahr, 2014, unpublished data).

Figure 3 Coral bleaching and recovery from 2014 event.

Photographs of the reef flat on the fringing reef surrounding Moku o Loʻe (Coconut Island) during the second large scale bleaching event in Kāneʻohe Bay in October 2014 and December 2014. Photographs by KD Bahr.

Lastly, these bleaching events displayed variation in interspecies bleaching susceptibilities. Bleaching and mortality responses have been shown to vary between individual corals, taxon, depth, and location (Grottoli, Rodrigues & Juarez, 2004). Coral species in similar habitats have shown different bleaching susceptibilities, indicating some coral species are more resistant than others to environmental stressors (Stimson, Sakai & Sembali, 2002). It has been suggested that the differences in coral susceptibility are linked to colony morphology, tissue thickness, and genetic constitution of the symbiotic algae (Rowan et al., 1997; Loya et al., 2001). As previously predicted (Hoegh-Guldberg, 1999; Loya et al., 2001), the thinner tissue corals (i.e., Pocillopora spp.) were the most susceptible to bleaching in the 1996 and 2014 bleaching events. Contrary to 1996, some colonies of M. capitata appeared to be more resilient to the 2014 bleaching conditions.

The Past, Present and Future of Coral Reefs in Kāneʻohe Bay

A timeline showing changes in condition of coral reefs (percent cover on shallow <2 m deep slopes of fringing and patch reefs) is presented as Fig. 4 based on the best available data. This timeline covers the “Polynesian Era” (1250 to 1778), the “Western Era” (1778 to present) and the “Future Era” (the present to 2040). Impacts during the Western Era were expanded into Fig. 5 in order to increase resolution of details.

Figure 4 Kāneʻohe Bay reef conditon over time.

Changes in coral reefs condition (percent coral cover) on shallow slopes (<2 m) of fringing and patch reefs during the Polynesian Era (1250–1778), the Western Era (1778–2015), and the Future Era (2015–2040). Percent coral cover during the Western Era is based on best available quantitative data. The Polynesian Era data is modified from Fig 4E in Kittinger et al., 2011. Future Era coral cover is estimated by the COMBO business as usual (red) (modified after Fig 7 in Buddemeier et al. (2008)) and by including coral adaptive responses (blue) RCP 6 with 60 year rolling climatology (modified after Fig. 3A in Logan et al. (2013)) in Kāneʻohe Bay.

Figure 5 Reef response to major impacts during Western Era.

Changes in percent coral cover in response to major impacts during the Western Era (1778–2015) based on best available data. A kite diagram weights influence of impact on the coral reef by thickness of the line over time. Perturbations of freshwater kills (blue) and bleaching events (red) occurrences are indicated by arrows in the timeline.

The polynesian era (from 1250–1778)

Kittinger et al. (2011) reconstructed ecological changes on Hawaiian reefs (Fig. 4) through an intensive review and assessment of archaeological deposits, ethnohistoric and anecdotal descriptions and modern ecological and fishery data. Social-ecological interactions in Hawaiian coral reef environments over the past 700 years were reconstructed using detailed datasets on ecological conditions, proximate anthropogenic stressor regimes and social change. They discovered previously undetected periods of recovery in Hawaiian coral reefs, including a historical recovery in the main Hawaiian Islands (MHI) between 1400 to 1820 AD and an ongoing recovery in the Northwest Hawaiian Islands (NWHI) from 1950 to 2009 + AD. These recovery periods have been attributed to a complex set of changes in underlying social systems, which served to release reefs from direct anthropogenic stressor regimes. Their results challenge conventional assumptions and reported findings that human impacts to ecosystems are cumulative and always lead to long-term trajectories of environmental decline. In contrast, recovery periods revealed that human societies have interacted sustainably with coral reef environments over long time periods, and that degraded ecosystems may still retain the resilience and adaptive capacity to recover from human impacts.

The western era (from 1778 to present)

Percent coral cover has declined since the beginning of the Western Era due to the suite of anthropogenic stressors and natural flood events described previously. Figure 5 presents a summary of changes in coral cover (% cover on shallow <2 m deep slopes of fringing and patch reefs). These estimates were developed using coral cover data and detailed observations (MacKay, 1915; Edmondson, 1929) along with surveys conducted after major impacts (Bosch, 1967; Maragos, 1972; Devaney et al., 1982; Fitzhardinge, 1985; Jokiel, 1991; Jokiel et al., 1993; Hunter & Evans, 1995; Stimson, Larned & Conklin, 2001; Jokiel & Brown, 2004; Bahr, Rodgers & Jokiel, 2015) and ongoing monitoring efforts (Jokiel & Brown, 2004; Rodgers et al., 2015; J Stimson, pers. comm., 2014). Entire reefs were removed by dredging with a large decrease in coral area. Figure 5 shows best estimates of recovery on the remaining reef surfaces on the reef slopes in shallow dredged areas (Maragos, 1972). Eutrophication from continuous sewage discharge from 1951 to 1979 increased productivity and promoted anoxic conditions in the south basin. In turn, these conditions reduced the coral cover greatly with continued sedimentation, floods and sewage discharge. Once the stressors were removed, rapid recovery in coral reef populations in the south basin occurred over a relatively short time period, approximately 20 years (Hunter & Evans, 1995).

Recovery from a freshwater kill in 1965 was delayed by sewage discharge and eutrophication in the south basin, but proceeded rapidly after sewage discharge was terminated in 1979. In contrast, rapid recovery from the 1988 flooding event (nine years after sewage discharge ended) was observed within 5–10 years (Jokiel et al., 1993). Overall coral cover has remained relatively stable since 2000, with half of the permanent transect sites surveyed by the Hawaii Coral Reef Assessment and Monitoring Program (CRAMP) showing an overall increase from 2000 to 2012 in coral cover (Rodgers et al., 2015). Minor fluctuations in coral cover may result from factors such as the 1996 bleaching event and unfavorable weather conditions including the prolonged period of rain and low irradiance between February and June 2006 (Stimson & Conklin, 2008). Low levels of coral mortality were observed from both the 1996 bleaching event (<2%) and the 2014 bleaching events (Jokiel & Brown, 2004; K Bahr, 2014, unpublished data) with recovery during the following winter in both cases. The primary exception to this generalization is that high mortality was observed during the 2014 event in an area that was previously impacted by a flood event that occur a few months prior (Bahr, Rodgers & Jokiel, 2015). The timeline in Figs. 4 and 5 appears to support this conclusion.

The future era (present to 2040)

The above discussion demonstrates the past resilience of Kāneʻohe Bay coral reef ecosystem to a broad array of severe environmental stressors; however, these reefs are now facing the unprecedented challenge of global climate change in addition to all these ongoing stressors. Two of the major aspects of climate change are increased sea surface temperatures and ocean acidification. The analysis of the global carbon budget (doi: 10.5194/essdd-7-521-2014) by the Carbon Dioxide Information Analysis Center (CDIAC), which is the primary climate-change data source, and information analysis center of the US Department of Energy (DOE) (http://cdiac.ornl.gov/) show unabated rapid global emissions of greenhouse gasses. Therefore, the global atmosphere and ocean will continue to warm in the coming years due to increased anthropogenic greenhouse gas production, causing an increase in the frequency and severity of bleaching events in the Hawaiian region. Moreover, the CDIAC and the Goddard Institute for Space Studies (GISS) report that 2014 was the warmest year on record; therefore, 15 of the warmest years on record have occurred since 1998 (Hansen et al., 2015). At the present time the use of the ‘business as usual’ (i.e., worst case) scenario is justified because recent global greenhouse gas emissions have exceeded the worst-case scenario and there is no evidence of a global shift to change that trend.

Conditions controlling coral bleaching, recovery and mortality in Hawaiian corals and corals throughout the world have been extensively described (reviewed by Brown, 1997; Hoegh-Guldberg, 1999; Jokiel, 2004; McClanahan et al., 2007). The first descriptions of coral thermal bleaching and recovery (Jokiel & Coles, 1974; Jokiel & Coles, 1977; Jokiel & Coles, 1990; Coles & Jokiel, 1977; Coles & Jokiel, 1978) have provided a solid basis for prediction when coupled with climate change models. By 1976, it was established that all the corals in the world were living within 1–2 °C of their upper temperature limit during the warmest summer months (Coles, Jokiel & Lewis, 1976; Coles & Brown, 2003). This observation has been verified by bleaching thresholds reported from throughout the world (Jokiel & Brown, 2004). The original work attracted little attention because at that time bleaching events were unknown. This changed with the first massive bleaching event off Panama in 1983, followed by more frequent and severe events leading to devastating impacts throughout the world during 1998 and 2005 (e.g., Glynn et al., 2001). Consequently, coral cover on reefs in the Caribbean has declined by 50% (Wilkinson, 2004). Hawaiʻi escaped these major events due to its location in the north central Pacific. The gradual rise in ocean temperature off Hawaiʻi shown in long-term records led Jokiel & Coles (1990) to predict that Hawaiian reefs were also approaching their upper thermal limits and that the first bleaching events were imminent. The prediction was verified by the 1996 bleaching event in Kāneʻohe Bay and subsequent events in the Northwest Hawaiian Islands in 2002 (Jokiel & Brown, 2004) with the awareness that more severe and more frequent bleaching events would follow. As predicted, a larger and more severe event occurred in 2014 (Neilson, 2014).

Quantitative data on Hawaiian coral response and models of future climate change (Houghton et al., 2007) have been used to construct mathematical models of how Hawaiian reefs will respond to future scenarios of global warming (Buddemeier et al., 2008; Hoeke et al., 2011). The Coral Mortality and Bleaching Output (COMBO) model (Buddemeier et al., 2008) uses a probabilistic assessment of the frequency of high temperature events under a future climate to address scientific uncertainties about potential adverse effects. Sensitivity analyses and simulation examples for Hawaiʻi demonstrate the relative importance of high temperature events, increased average temperature, and increased CO2 concentration on the future status of coral reefs. The results of the COMBO modeling indicate that Kāneʻohe Bay will show a more rapid rate of coral loss due to global warming compared to exposed ocean reefs with transitions occurring 5–10 years earlier. The general pattern of more severe bleaching in embayments and shallow areas with poor water exchange was upheld by observations of the 2014 bleaching event (Neilson, 2014) (Figs. 4 and 5). Results of these Kāneʻohe Bay “worst case” simulations to the year 2040 are consistent with results of previous studies (Hoegh-Guldberg, 1999; Sheppard, 2003) from throughout the world in demonstrating the potential for unprecedented levels of future coral reef decline.

Conversely, other simulations that take into account ecological and adaptive processes suggest that there may be capacity for coral communities to buffer these negative declines. Wooldridge et al. (2005) developed a prototype decision-support tool, called ‘ReefState,’ which integrates the outcomes of management interventions within a ‘belief network’ of connected variables that describe future warming, coral damage and coral recovery. In the inshore waters of the central Great Barrier Reef, Australia, their worst case scenarios suggest that reefs will become devoid of significant coral cover and associated biodiversity by 2050. Even under more optimistic (low) rates of future warming, the persistence of hard coral dominated reefs beyond 2050 will be heavily reliant on the ability of corals to increase their upper thermal bleaching limits by ∼0.1 °C per decade, and management actions that produce local conditions that constrain algal biomass proliferation during inter-disturbance intervals. Donner, Knutson & Oppenheimer (2007) adapted the NOAA Coral Reef Watch bleaching prediction method to the output of a low- and high-climate sensitivity General Circulation Model (GCM). They developed and tested algorithms for predicting mass coral bleaching with GCM-resolution sea surface temperatures for thousands of coral reefs, using a global coral reef map and 1985–2002 bleaching prediction data. These algorithms were used to determine the frequency of coral bleaching and required thermal adaptation by corals and their endosymbionts. Data from this model led them to conclude that bleaching could become an annual or biannual event for the vast majority of the world’s coral reefs in the next 30–50 years.

As in the case of other coral reef modelers (Wooldridge et al., 2005; Buddemeier et al., 2008; Logan et al., 2013) they emphasize the importance of adaptation in delaying mass bleaching events and associated coral mortality. Buddemeier et al. (2008) found that even under conditions of moderate warming the total adaptation required by most reefs may exceed 2 °C in the latter half of the century. Possibilities of adaptive processes (e.g., genetic adaptation, acclimatization, and symbiont shuffling) to thermal stress may influence the bleaching threshold and frequency of bleaching events of various coral species. In the absence of adaptive processes, the NOAA Coral Reef Watch bleaching prediction method was determined to over predict the present day bleaching frequency (Logan et al., 2012). This suggests that corals may have adapted to some of the increases in SST over the industrial period (Logan et al., 2013). At the other end of the spectrum, Logan et al. (2013) predict less than 3% of the reefs of the world will experience frequent bleaching by mid-century with a 100 year window of adaptation and with a 10 year temporary threshold response. Nonetheless, necessary observations and empirical data have not yet validated a model that accounts for adaptive processes (Logan et al., 2013). Future research is needed to test the rate and limit of different adaptive responses for coral species.

Berkelmans (2002) constructed a predicted bleaching-response model from high-resolution in situ temperature records and historical observations of coral bleaching throughout the Great Barrier Reef. Distinct spatial trends exist in the thermal sensitivity of coral populations that correspond with location. This suggests that considerable thermal adaptation has taken place over small (10 s of km) and large (100 s to 1000 s of km) spatial scales. Likewise, Buddemeier et al. (2008) found that Kaneohe Bay baseline temperature produced a more realistic bleaching estimate than the oceanic temperature data suggested that some local adaptation may be occurring, although not enough to escape eventual bleaching.

Advanced modeling efforts suggests it is extremely unlikely that viable coral populations will exist in the shallow waters of the Hawaiian Archipelago in 2100 (Hoeke et al., 2011). However, these model outcomes were highly sensitive to increasing the tolerance to future levels of heat stress. Corals will fare much better if they can adapt to episodic mortality through processes such as selection of more thermally tolerant algal symbionts (Baker, Glynn & Riegl, 2008), or taxonomic succession of more resistant or resilient genera (Baird & Maynard, 2008). In the Hoeke et al. (2011) modeling study, potential adaptation was the single most sensitive parameter. If corals can increase their threshold for heat stress at 0.1 °C/decade, the model suggests a decline of 25% to 75% (rather than 100%) in coral cover for most locations by the end of the century. Many of the models use sea surface temperature only. Modeling with temperature from deeper water showed that that even in the “less resilient” case (no ability of corals to adapt to higher temperature), areas of viable coral reefs can persist on deeper fore reefs or in areas where upwelling of cooler water is occurring.

The corals in Kāneʻohe Bay are currently living at temperatures and acidification regimes that will not be experienced for decades on open coastal reefs across the Hawaiian Archipelago. Does this make these corals more susceptible or more resilient to future climate conditions? The COMBO and adaptive response models make very different predictions about the future of coral reefs in the bay (Fig. 4). Despite the fact that reefs in the bay currently persist under these conditions, the limits of Kāneʻohe Bay reef resilience in the face of global climate change remain a major focus for future research.

Conclusions

Corals in Kāneʻohe Bay have recovered from major human impact (i.e., long-term harvest, sewage, sedimentation, etc.) as well as major natural disturbances (i.e., fresh water kills, etc.). Recovery from natural perturbations tends to occur on the scale of 5–20 years in Kāneʻohe Bay, but can be prevented by presence of chronic anthropogenic stressors (Jokiel et al., 1993). Thus, future recovery and persistence of these reefs will require continued attention to local pollution, sedimentation and harvest issues. Kāneʻohe Bay is now faced with the ultimate anthropogenic stress of global climate change. The reefs of Kāneʻohe Bay have shown remarkable resilience to a wide variety of natural and anthropogenic insults over the centuries, but the pressing new question centers on whether coral reefs can survive continuously increasing temperature and ocean acidification which will be punctuated by a series of perturbations including bleaching events and fresh water kills. One aspect of this question is whether or not recovery from these events can occur under conditions of increasing temperature and increasing ocean acidification along with changes in sea level, precipitation and more severe storm activity predicted under climate change models. Local stressors can be diminished, but climate change stressors will continue and are only expected to increase with time.

Anthony et al. (2015) proposed an operational framework for identifying effective actions that enhance resilience and support management decisions while reducing reef vulnerability. They proposed an adaptive resilience-based management (ARBM) framework based on biological and ecological processes that drive resilience of coral reefs in different environmental and socio-economic settings, and suggested a set of guidelines for how and where resilience can be enhanced via management interventions. However, they clearly state that: “As climate change and ocean acidification erode the resilience and increase the vulnerability of coral reefs globally, successful adaptive management of coral reefs will become increasingly difficult.”

Thus, among the most pressing questions facing the future of coral reefs is what is the capacity for adaptation and how long might it take? Even if adaptation can occur, can it happen quickly enough to matter for the future of coral reefs? Whatever the answers are to these questions, the biological responses of the system are critical to understand, because ecosystem feedbacks have a much greater effect than average conditions on seawater carbonate chemistry (Jury et al., 2013). The response of coral reef communities to future environmental conditions is a topic of enormous concern and considerable debate. Changes in coral community structure and extinctions have been linked with extreme climate events in the past (reviewed by Budd, 2000; Stanley, 2003; Mora et al., 2013), however little information is available concerning the biological response to the interactions between stressors on modern reefs. Therefore, our ability to predict future climate changes on coral reefs remains limited, and this is another critical area for future research and better understanding. The well-documented effects of anthropogenic and natural stressors on the Kāneʻohe Bay reef ecosystem may provide insights to these questions as well as allow this region to serve as an exemplary research site for future work on climate change.

Supplemental Information

Supplemental Information 1 Detailed descriptions of Kāneʻohe Bay ecosystem

Extensive detail of the description of the features, physical, biological and chemical characteristics of Kāneʻohe Bay.

Click here for additional data file.

Additional Information and Declarations

Competing Interests

Author Contributions

Robert Toonen is an Academic Editor for PeerJ.

Keisha D. Bahr conceived and designed the experiments, analyzed the data, contributed reagents/materials/analysis tools, wrote the paper, prepared figures and/or tables, reviewed drafts of the paper.

Paul L. Jokiel conceived and designed the experiments, analyzed the data, contributed reagents/materials/analysis tools, wrote the paper, reviewed drafts of the paper.

Robert J. Toonen conceived and designed the experiments, contributed reagents/materials/analysis tools, wrote the paper, reviewed drafts of the paper.

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
