# Peer review of "The unnatural history of Kāne‘ohe Bay: coral reef resilience in the face of centuries of anthropogenic impacts"

_PeerJ, doi:10.7717/peerj.950_

## Round 0.1 · original submission · Major Revisions

I believe the reviewers have given you some good / fair feedback and numerous suggestions to improve the manuscript.

Reviewer 1 ·

Basic reporting

Review of Bahr et al. PeerJ
General Comments:
I was excited to read this paper as I think the Kane’ohe Bay coral reef story is an interesting one with a complex background. That being said the manuscript is overly long and reads more like a student’s term paper than a scientific publication. There are many sections that are very general and not specific to Kane’ohe Bay that should be removed--almost every section starts off with a general paragraph that is not specific to Kane’ohe Bay. In general, I think it would help if the authors define their primary goal in writing this paper up front as I think this may help to narrow the focus and make a more clear paper. The authors provide extensive detail of the history of Hawaiian settlement to contextualize the types of “impact” that the reefs in the bay have experienced but these sections read more like something out of a history book than sections out of a review paper. My main recommendation for the authors is that they really focus the paper more on the data that exists on a) the impacts to the reefs and any evidence for how these impacts have (or have not) affected the health of the reefs over time. I would recommend that the authors cut the text by half and focus solely on the Kane’ohe Bay story focusing on the history, science, impacts, current status, etc.

Specific Comments:
-In reading between the lines, it seems from the abstract that the primary goal of the authors is to describe if/how the health of the coral reefs in the bay has changed over time. However, in reading the manuscript the authors summarize the context in which the bay exists in extensive detail. They then describe “Kane’ohe Bay Coral” which actually includes general information on corals in Hawaii, and very general information on corals in general (e.g., summary paragraphs on how corals reproduce, what coral bleaching is, what factors are known to affect coral morphologies, etc.). The paper would benefit from a major focusing of these sections to tell the Kane'ohe story.
-The section on historical events affecting K-Bay corals is good but could also be refined and shorted. Some sections (such as the one on invasive species) are quite general and not focused on K-Bay in particular. There is no discussion of invasive invertebrates or fish which I believe exist in the bay as well.
-There is surprisingly no discussion of overfishing or impacts of fishing (gillnets, etc.) on the reef communities in the bay. Clearly this is an important “impact”.
-The authors provide no data on or evidence for their statements about the reefs in Kane’ohe Bay being healthier than many reefs around the world. What is the basis for their statements? How do they define a healthy reef? Can they synthesize data on how coral cover has changed over time (if this is the metric they are most interested in)? In fact there is no real summary of a) what the reef communities of the bay look like now or b) how they have changed over time. I also find it a bit naïve to only consider corals as the primary indicator of reef condition. What about CCA, turf and macroalgae or other invertebrates?
-In general I think this could be a really interesting review but as written now there is just too much ancillary information. I recommend that the authors try to trim off the general text and tell the Kane'ohe specific story.

Experimental design

The authors need to outline the goals of the review more clearly.

Validity of the findings

Without clearly defining the goals of the paper it is difficult to evaluate the findings. The authors certainly need to provide more information on how they come to their conclusions.

Additional comments

see above sections

Reviewer 2 ·

Basic reporting

Reference on Line 180 is missing a year.
The information beginning on line 564 and continuing until the end of the paragraph, should have a citation.
There are both metric and imperial measures of area and distance. Feet and square miles should be converted to metric.
Is line 1149 the full reference?
In the abstract, perhaps "discovery" by Cook might be rephrased to something like "arrival by the first Europeans".
In the abstract, can fish pond enclosures be "personified" by something. Strange word choice.
Line 84: Consider including a tidal range. Is it .6m?

Experimental design

As this is a review paper, there is no hypothesis or experimental design. However, there are elements of the structure of the paper that could be improved. For example, the 1996 bleaching event is reintroduced and reviewed numerous times with varying levels of detail. The authors might consider reorganizing the paper so that that discussion is only treated once (and perhaps again in the conclusions). In addition, there are several paragraphs of anthropology and coral biology that are completely disconnected from the discussion of the bay. The history of the surrounding area is certainly interesting but for the purposes of the this paper (and its stated goals) this discussion should be connected to specific impacts to the bay and the coral community therein. In another example, lines 354-364 contain an accurate discussion of factors that influence coral morphology but they are totally unconnected to how those factors vary in the bay. In that way, the paper risks being a giant grab bag of information that isn't explicitly connected to impacts on the corals of Kane ohe' bay (which is the central concern of the title). I would advise a careful edit that either eliminates unconnected tidbits of information or makes their connection to the corals of the bay more explicit.

Line 426 is missing the word "to"
I think there should be a space between lines 629 and 620
Line 634: Is particulars supposed to be particles?

Validity of the findings

I think all of the statements contained in this review are supported by the cited works.

See above: There are several paragraphs which are not explicitly connected to the central topic of the review.

Additional comments

This paper is a very thorough review of the abiotic and biotic impacts to the bay. It will serve as an excellent reference work for PIs and graduate students in the decades to come! I made suggestions that might "tighten-up" and improve the manuscript.

---

## Round 0.2 · accepted · Accept

Thank you for making so many edits in response to the reviewer suggestions. I now love this paper. It gives a really good sense of the natural history of the bay and explains the temporal patterns and some of the drivers of change over the last several decades and beyond. Until now I was pretty confused about the current state of the reefs in the bay and was surprised to learn how well they've recovered from earlier disturbances. It is pretty amazing how resilient such a low diversity reef, in such close proximity to people (and lots of them!) can be. It reminds me of both our work in Jamaica and the findings of Zhang et al in PeerJ (no relationship between coral richness and resilience).